# CAMVR: Context-Adaptive Multi-View Representation Learning for Dense Retrieval

## Abstract

The recently proposed MVR (Multi-View Representation) model achieves remarkable performance in open-domain dense retrieval. In MVR, the document can match with multi-view queries by encoding the document into multiple representations. However, these representations tend to collapse into the same one when the percentage of documents answering multiple queries in training data is low. In this paper, we propose a CAMVR (Context-Adaptive Multi-View Representation) learning framework, which explicitly avoids the collapse problem by aligning each viewer token with different document snippets. In CAMVR, each viewer token is placed before each snippet to capture the local and global information with the consideration that answers of different view queries may scatter in one document. In addition, the view of the snippet containing the answer is used to explicitly supervise the learning process, from which the interpretability of view representation is provided. The extensive experiments show that CAMVR outperforms the existing models and achieves state-of-the-art results.

## 1 Introduction

Dense retrieval approaches based on pre-trained language models (Devlin et al., 2019; Liu et al., 2019) achieve significant retrieval improvements compared with sparse bag-of-words representation approaches (Jones, 1972; Robertson et al., 2009). A typical dense retriever usually encodes the document and query into two separate vector-based representations through a dual encoder architecture (Karpukhin et al., 2020; Lee et al., 2019; Qu et al., 2021; Xiong et al., 2020), then relevant documents are selected according to the similarity scores between their representations. To this end, the major approaches improve the retrieval performance by learning a high-quality document representation, including hard negative mining (Zhan et al., 2021; Xiong et al., 2020; Qu et al., 2021) and task-specific pre-training (Gao & Callan, 2021b; Oguz et al., 2022).

Intuitively, the capacity of a single-vector representation is limited (Luan et al., 2020) when the document is long and corresponds to multi-view queries. Recently, Zhang et al. (2022) proposes a MVR (Multi-View Representation) model to improve the representation capacity by encoding a document into multiple representations, and the similarity score between a query and document is determined as the maximum score calculated with their dense representations. However, when the percentage of documents answering multiple queries in training data is low, the multiple vector-based representations of each document tend to collapse into the same one. For brevity, we use the average number of queries (AVQ) per document to represent the percentage of documents answering multiple queries in training data. As show in Table 2, when the AVQ value is small (1.0), multi-view representations collapse into the same one and the performance deteriorates sharply in MVR.

Considering that the answers of queries may scatter in different snippets of one document and the viewer tokens in MVR model are located in front of the document, this makes it difficult to perceive the information from the snippets that may contain the scattered answers. As shown in Figure 1, the two answers are located in the 2nd and 4th snippets in example 1, while the two answers are located in the 5th and 6th snippets in example 2. Obviously, MVR will not be able to adaptively capture the different answer information because the positions of viewer tokens are fixed. Well-learned multi-view representations are expected to capture the information from different snippets. In this paper, we propose a CAMVR (Context-Adaptive Multi-View Representation) learning framework, which explicitly avoids the collapse problem by aligning viewer tokens with snippets (as illustrated

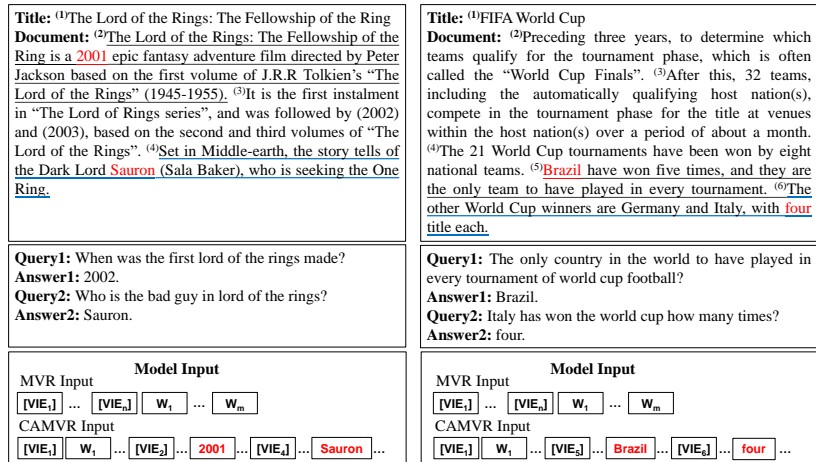

Figure 1: Two examples from Natural Question Dataset and the input comparison of MVR and CAMVR. $[VIE_i]$ is a viewer token and the answers are marked red. Compared with MVR model where the viewer tokens are located before the document, our CAMVR model is able to adaptively capture the information of answer snippets by aligning each viewer token with each snippet.

in Figure 1). Firstly, to allow views to adaptively capture the information of answer snippets in a document, we locate each viewer token before each snippet. Therefore, each view representation is able to aggregate local and global information via the self-attention mechanism of transformer (Vaswani et al., 2017). Furthermore, we use the view corresponding to the answer snippet of a given query to explicitly supervise the learning task, which allows the view to attend to the corresponding local snippet, thereby increasing the view interpretability.

The contributions of this paper are as follows:

- Different from MVR model, where the representations would collapse into the same one when the percentage of documents answering multiple queries in training data is low, our proposed CAMVR learning framework is able to adaptively align each viewer token with each document snippet to capture fine-grained information of local snippets and global context, which effectively avoid the collapse problem.

- We specify the answer snippet for positive document to supervise the learning process and allows the representation to attend to local snippets, which provides the interpretability for each view representation.

- Extensive experiments are conducted to evaluate our proposed CAMVR in terms of the analysis of overall retrieval performance, collapse problem, view interpretability and view number, etc. The experimental results on open-domain retrieval datasets show the effectiveness of our proposed model, which achieves the state-of-the-art performance.

## 2 RELATED WORK

In this section, we review the existing three model architectures for dense retrieval: dual encoder, late interaction and multi-vector model respectively. As shown in Figure 2, late interaction and multi-vector model are the variants of dual encoder.

Dual encoder (Karpukhin et al., 2020) (Figure 2(a)) is widely used in the first-stage document retrieval to obtain the relevant documents from a large-scale corpus. In general, the pre-trained model BERT (Devlin et al., 2019) is utilized as a base model for dual encoder considering its text representation ability learned from massive data. To improve retrieval performance, Lee et al. (2019) and Chang et al. (2020) further pre-train the BERT with inverse cloze task, body first selection and wiki link prediction. To make the language models be ready for dense retrieval tasks, Gao & Callan (2021a;b) propose pre-training architectures coCondenser and Condenser, where a contrastive learn-

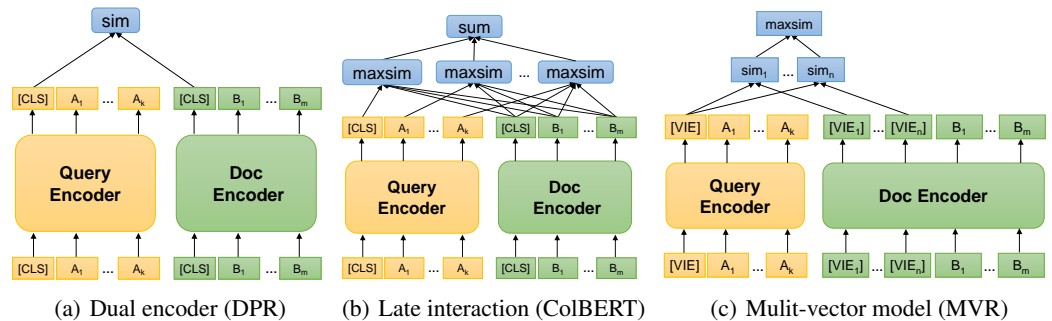

Figure 2: The comparison of different model architectures designed for dense retrieval.

ing task is additional considered in coCondenser. Besides pre-training, negative sampling is adopted to improve retrieval performance by mining high-quality negatives. DPR (Karpukhin et al., 2020) picks out hard negatives through the BM25 retriever. Yang et al. (2017) and Xiong et al. (2020) generate hard negatives dynamically with the newest checkpoint in training process. Qu et al. (2021) and Ren et al. (2021) use cross encoder to mine hard negatives.

The interaction between each query and document is also used to improve retrieval performance by exploiting their token-level relevance (Nogueira & Cho, 2019; MacAvaney et al., 2019; Dai & Callan, 2019; Jiang et al., 2020). However, such token-level interaction is computation intensive, which is impractical for the first-stage retrieval given a large-scale document collection. Therefore, late interaction paradigm (Figure 2(b)) is proposed to improve the computational efficiency. In general, each pair of query and document is encoded independently and the similarity between them is computed through a fine-grained interaction mechanism. In ColBERT (Khattab & Zaharia, 2020; Santhanam et al., 2022), a relevance score is determined as the sum of the maximum similarity between the tokens of the query and document. Poly-Encoder (Humeau et al., 2020) provides an attention mechanism to learn the global features of a document, which is more efficient than directly using all token-level features of the document. COIL (Gao et al., 2021) is proposed to only calculates the scores of the overlapped tokens between a query and document to reduce the computational costs.

Since the representation capacity of single vector is limited for a long document and late interaction between all tokens still requires high computational costs, multi-vector models (Figure 2(c)) are therefore proposed. ME-BERT (Luan et al., 2020) is proposed to improve retrieval performance by using the first $m$ token embeddings as the document representation. However, this would lose useful information in the latter part of the document. DRPQ (Tang et al., 2021) encodes a document with class centroids by clustering all token embeddings. MVR (Zhang et al., 2022) encodes a document into multi-view representations and each of them can be used to answer different view queries. However, the multi-view representations would collapse into the same one when the percentage of documents answering multiple queries in training data is low. Though DCSR (Hong et al., 2022) produces multiple vectors for the document, there is a gap between the training and inference tasks, which limits its retrieval performance.

## 3 METHOD

### 3.1 PRELIMINARY

Dual encoder is widely used to encode a query (by the query encoder) and document (by the document encoder) into a single vector respectively in open-domain dense retrieval as shown in Figure 2(a). Given a query $\boldsymbol{q}$ and a document $\boldsymbol{d}$, the similarity score function is determined as follows:

$$f(\boldsymbol{q}, \boldsymbol{d}) = \text{sim}(E_Q(\boldsymbol{q}), E_D(\boldsymbol{d})), \tag{1}$$

where $E_Q(\cdot)$ is the query encoder, $E_D(\cdot)$ is the document encoder and $\mathrm{sim}(\cdot)$ is the inner product function. Then, the loss function of dual encoder is defined as

$$\mathcal{L} = -\log \frac{\exp(f(\boldsymbol{q}, \boldsymbol{d}^+)/\tau)}{\exp(f(\boldsymbol{q}, \boldsymbol{d}^+)/\tau) + \sum_l \exp(f(\boldsymbol{q}, \boldsymbol{d}_l^-)/\tau)}, \tag{2}$$

where $\boldsymbol{d}^+$ and $\boldsymbol{d}_l^-$ represent the positive and negative documents for $\boldsymbol{q}$, and the hyper-parameter $\tau$ is a scaling factor for model optimization (Sachan et al., 2021).

MVR model takes the dual encoder architecture to encode the query and document into vectors, as shown in Figure 2(c). For the query encoder, the $[CLS]$ token is replaced with a viewer token $[VIE]$ to produce the query representation. Formally, the query representation is given as

$$E(\boldsymbol{q}) = E_Q([VIE] \circ \boldsymbol{q} \circ [SEP]), \tag{3}$$

where $E(\boldsymbol{q})$ is determined by the vector of viewer token $[VIE]$, $[SEP]$ is a special token in BERT and $\circ$ is the concatenation operation. For the document encoder, viewer tokens $[VIE_i](i = 1, 2, \ldots, n)$ are placed at the beginning of document to produce multi-view representations:

$$E_1(\boldsymbol{d}), E_2(\boldsymbol{d}), \ldots, E_n(\boldsymbol{d}) = E_D([VIE_1] \ldots [VIE_n] \circ \boldsymbol{d} \circ [SEP]), \tag{4}$$

where the i-th view representation $E_i(\boldsymbol{d})$ of document is determined by the vector of $[VIE_i]$. The similarity score function between the query $\boldsymbol{q}$ and i-th view of the document is defined as

$$f_i(\boldsymbol{q}, \boldsymbol{d}) = \mathrm{sim}(E(\boldsymbol{q}), E_i(\boldsymbol{d})), \tag{5}$$

where $\mathrm{sim}(\cdot)$ is the inner product function. The overall similarity score of query $\boldsymbol{q}$ and document $\boldsymbol{d}$ is aggregated by

$$f(\boldsymbol{q}, \boldsymbol{d}) = \max_i \{ f_i(\boldsymbol{q}, \boldsymbol{d}) \}. \tag{6}$$

Different from the loss function of dual encoder, MVR further adds a regularization for different views to alleviate the collapse problem. Formally, the final loss function is given as

$$\mathcal{L} = -\log \frac{\exp(f(\boldsymbol{q}, \boldsymbol{d}^+)/\tau)}{\exp(f(\boldsymbol{q}, \boldsymbol{d}^+)/\tau) + \sum_l \exp(f(\boldsymbol{q}, \boldsymbol{d}_l^-)/\tau)} - \lambda \log \frac{\exp(f(\boldsymbol{q}, \boldsymbol{d}^+)/\tau)}{\sum_i \exp(f_i(\boldsymbol{q}, \boldsymbol{d}^+)/\tau)}, \tag{7}$$

where $\lambda$ is a hyper-parameter.

## 3.2 CONTEXT-ADAPTIVE MULTI-VIEW REPRESENTATION

In MVR model, the input of a viewer token is given as:

$$\boldsymbol{e}_i = \boldsymbol{v}_i + \boldsymbol{p}_0 + \boldsymbol{o}, \quad i = 1, \ldots, n \tag{8}$$

where $n$ is the number of viewer tokens, $\boldsymbol{v}_i$ is the token embedding of $[VIE_i]$, $\boldsymbol{p}_0$ and $\boldsymbol{o}$ are the position and segment embeddings, respectively. Since $\boldsymbol{p}_0$ and $\boldsymbol{o}$ are not altered for all viewer tokens, Equation (8) suggests that all the differences in view representations are caused from the token embedding. From another perspective, Equation (8) can be equivalently rewritten as follows:

$$\boldsymbol{e}_i = \tilde{\boldsymbol{v}}_0 + \tilde{\boldsymbol{p}}_i + \boldsymbol{o}, \quad i = 1, \ldots, n \tag{9}$$

where $\tilde{\boldsymbol{v}}_0 = \boldsymbol{p}_0$ and $\tilde{\boldsymbol{p}}_i = \boldsymbol{v}_i$ for $i = 1, \ldots, n$. That is, the viewer token embedding $\tilde{\boldsymbol{v}}_0$ is fixed, while the position embedding is varying. However, as shown in Figure 1, answer snippets for different queries are scattered in a document. The views with $n$ fixed position embeddings in Equation (9) are not able to capture the information of fine-grained scattered answer snippets for each document. When the percentage of documents answering multiple queries in training data is low, the information to align the document views to different queries is insufficient. Thus, the multi-view representations tend to collapse into the same one and the performance will deteriorate sharply.

In this paper, we propose a CAMVR (Context-Adaptive Multi-View Representation) learning framework in which the position embeddings of viewer tokens are adaptively changed with the context of a document. As shown in Figure 1, it is crucial for different views to perceive fine-grained information in different answer snippets. Since each document snippet may answer some queries, we place a viewer token before each snippet, aiming to capture the information of local snippet and global context.

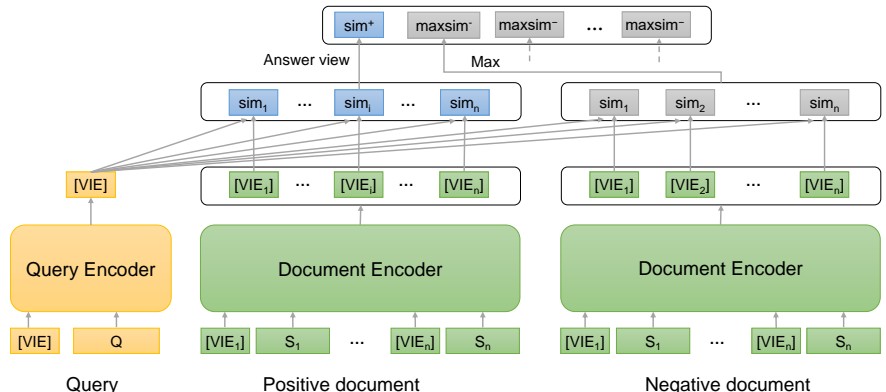

Figure 3: The architecture of context-adaptive multi-view representation learning framework. $\text{sim}^+$ denotes the similarity score between the query representation and answer view representation of positive document. The $\text{maxsim}^-$ denotes the similarity score between the query and one negative document.

---

**Algorithm 1** Document Splitting

---

**Input**: A document $\boldsymbol{d}$.
**Parameter**: View number $n$ and empty sentence $\boldsymbol{b}$.
**Output**: A sequence $S$ containing $n$ snippets.
1: Split $\boldsymbol{d}$ into $k$ sentences $S = [\boldsymbol{s}_1, \boldsymbol{s}_2, \ldots, \boldsymbol{s}_k]$ by the $sent\_tokenize$ function in NLTK toolkit.
2: **if** $k \leq n$ **then**
3:     Add $n$-$k$ empty sentences $\boldsymbol{b}$ to $S$ and obtain the final snippets $S = [\boldsymbol{s}_1, \boldsymbol{s}_2, \ldots, \boldsymbol{s}_k, \boldsymbol{b}, \ldots, \boldsymbol{b}]$.
4: **else**
5:     **while** $len(S) > n$ **do**
6:         Identify the shortest sentence $\boldsymbol{s}_i$ in $S$.
7:         Merge $\boldsymbol{s}_i$ with its shorter adjacent sentence $\boldsymbol{s}_j$ into a new snippet $\boldsymbol{s}_i'$.
8:         $\boldsymbol{s}_i \leftarrow \boldsymbol{s}_i'$.
9:         Remove $\boldsymbol{s}_j$ from $S$.
10:     **end while**
11: **end if**

---

To guarantee the sentence completeness of each snippet, we design a heuristic method to partition the document into snippets, and the details are described in Algorithm 1. For the document $\boldsymbol{d}$, we can obtain the $n$ snippets $[\boldsymbol{s}_1, \ldots, \boldsymbol{s}_n]$ with the document splitting algorithm. Then the input context of query and document can be given as follows:

$$\tilde{\boldsymbol{q}}: \quad [VIE] \circ \boldsymbol{q} \circ [SEP],$$
$$\tilde{\boldsymbol{d}}: \quad [VIE_1] \circ \boldsymbol{s}_1 \cdots [VIE_n] \circ \boldsymbol{s}_n \circ [SEP].$$

We treat viewer tokens and document tokens equally in position embedding, and let the viewer token embedding be the same with that of MVR model. Thus, the input of $[VIE_i]$ is given as

$$\boldsymbol{e}_i = \boldsymbol{v}_i + \boldsymbol{p}_{a_i} + \boldsymbol{o}, \qquad i = 1, \ldots, n; \ a_i \in [0, L] \tag{10}$$

where $a_i$ is the i-th viewer position in the sequence and $L$ is the maximum sequence length. The position embedding is same with that of vanilla pre-trained model, mainly considering its capability in perceiving local and global information which is proven in MLM (Masked Language Model) task. In addition, different viewer token embeddings help to distinguish adjacent views, making their representations complementary and avoiding the collapse problem.

We use the last hidden states of dual encoder as the representation of the query and document,

$$E(\boldsymbol{q}) = E_Q([VIE] \circ \boldsymbol{q} \circ [SEP]), \tag{11}$$
$$E_1(\boldsymbol{d}), E_2(\boldsymbol{d}), \ldots, E_n(\boldsymbol{d}) = E_D([VIE_1] \circ \boldsymbol{s}_1 \cdots [VIE_n] \circ \boldsymbol{s}_n \circ [SEP]), \tag{12}$$

where $E(\boldsymbol{q})$ is the query representation, and $E_i(\boldsymbol{d})(i = 1, 2, \ldots, n)$ is i-th view representation of the document. The definition of similarity function for query and document refers to MVR model.

In our method, each view representation contains the information of local snippets and global context. To make it attend more to local snippets, we specify the view of answer snippet to align with the query in training process, as shown in Figure 3. Let the i-th view be the answer view, then the loss function of CAMVR is defined as follows:

$$\mathcal{L} = -\log \frac{\exp(f_i(\boldsymbol{q}, \boldsymbol{d}^+)/\tau)}{\exp(f_i(\boldsymbol{q}, \boldsymbol{d}^+)/\tau) + \sum_l \exp(f(\boldsymbol{q}, \boldsymbol{d}_l^-)/\tau)}, \tag{13}$$

where $f_i(\boldsymbol{q}, \boldsymbol{d}^+)$ is the similarity score between the query and answer view of positive document, and used as the similarity score between the query and positive document in training process.

In inference, the similarity score is given by maximizing over the similarity scores between the query representation and document multi-view representations. We firstly use the document encoder of our model to encode all documents in the retrieval corpus into multi-view representations and build the index for each document view. Then we retrieve the relevant documents for the given query, and boost the retrieval process by the ANN (Approximate Nearest Neighbor) technique (Johnson et al., 2019). In contrast, the time complexity of CAMVR is the same with MVR, and lower than other multi-vector models such as DRPQ (Tang et al., 2021) and DSCR (Hong et al., 2022).

## 4 EXPERIMENTS

### 4.1 DATASETS

**SQuAD** (Rajpurkar et al., 2016) is a crowdsourced reading comprehension dataset. The version we used in this paper is created by DPR (Karpukhin et al., 2020) and it contains 70k training data. Since the test data is not released, we perform the retrieval evaluations in terms of top 5/20/100 accuracy on the development set.

**Natural Question** (NQ) (Kwiatkowski et al., 2019) is a large dataset for open-domain QA. All the queries are scraped from the Google search engine and anonymized. The documents are collected from Wikipedia articles. According to DPR (Karpukhin et al., 2020), 60k training data of NQ is used.

**TriviaQA** (Joshi et al., 2017) is a reading comprehension dataset containing over 650k query-answer-document triples. All the documents are collected from Wikipedia and Web. In our experiments, we leverage the version released by DPR (Karpukhin et al., 2020) which contains 60k training data.

The retrieval corpus used in our experiments contains 21,015,324 documents. According to DPR (Karpukhin et al., 2020), all the documents are non-overlapping chunks of 100 words.

### 4.2 IMPLEMENTATION DETAILS

We use 2 NVIDIA Tesla A100 GPUs (with 40G RAM) to train MVR and CAMVR models with a batch size of 56 on each GPU. To save the computational resources, the length of query and document are set to 128 and 256, respectively. In addition, automatic mixed precision and gradient checkpoints (Chen et al., 2016) are used. We use the Adam optimizer with a learning rate of 1e-5, and the dropout rate is set to 0.1. Other hyperparameter settings follow Zhang et al. (2022). To make a fair comparison with baseline models, we follow the mined hard negatives strategy adopted in coCondenser (Gao & Callan, 2021a) and MVR (Zhang et al., 2022) in Section 4.3. Note that we use the pre-trained coCondenser as the base model without any further pre-training.

### 4.3 RETRIEVAL PERFORMANCE

We compare our CAMVR model with the previous state-of-the-art models. These models can be divided into two categories based on the representations of documents: single-vector models and multi-vector models. The single-vector models we compare in our experiments include ANCE (Xiong et al., 2020), RocketQA (Qu et al., 2021), Condenser (Gao & Callan, 2021b), DPR-PAQ

(Oguz et al., 2022) and coCondenser (Gao & Callan, 2021a). For multi-vector models, we consider DRPQ (Tang et al., 2021) and MVR (Zhang et al., 2022) models for comparison.

As the experimental results shown in Table 1, our CAMVR model outperforms all the existing models. Specifically, our CAMVR achieves 0.7 (on SQuAD), 0.4 (on NQ), and 3.0 (on Trivia QA) points of top5 accuracy improvements compared with the performance of MVR model. The improvement is more significant on Trivia QA. The reason is that the AVQ value of the Trivia QA (1.2) is too small, as a result, the performance of MVR dercreases since the multi-view representations tend to collapse into the same one. In contrast, the multi-view representations of our CAMVR model can capture the information of local snippet and global context. In conclusion, the improvements on SQuAD, NQ and Triva QA datasets suggest that our CAMVR model is able to perform well for different AVQ values, especially for a small AVQ value.

| Method | SQuAD | | | Natural Question | | | Trivia QA | | |
|---|---|---|---|---|---|---|---|---|---|
| | R@5 | R@20 | R@100 | R@5 | R@20 | R@100 | R@5 | R@20 | R@100 |
| BM25 | - | - | - | - | 59.1 | 73.7 | - | 66.9 | 76.7 |
| DPR | - | 76.4 | 84.8 | - | 74.4 | 85.3 | - | 79.3 | 84.9 |
| ANCE | - | - | - | - | 81.9 | 87.5 | - | 80.3 | 85.3 |
| RocketQA | - | - | - | 74.0 | 82.7 | 88.5 | - | - | - |
| Condenser | - | - | - | - | 83.2 | 88.4 | - | 81.9 | 86.2 |
| DPR-PAQ | - | - | - | 74.5 | 83.7 | 88.6 | - | - | - |
| DRPQ | - | 80.5 | 88.6 | - | 82.3 | 88.2 | - | 80.5 | 85.8 |
| coCondenser | - | - | - | 75.8 | 84.3 | 89.0 | 76.8 | 83.2 | 87.3 |
| coCondenser | 73.2 | 81.8 | 88.7 | 75.4 | 84.1 | 88.8 | 76.4 | 82.7 | 86.8 |
| MVR | 76.4 | 84.2 | **89.8** | 76.2 | 84.8 | 89.3 | 77.1 | 83.4 | 87.4 |
| **CAMVR** | **77.1** | **84.5** | 88.6 | **76.6** | **85.1** | **89.9** | **80.1** | **85.5** | **88.8** |

Table 1: Retrieval performance on SQuAD dev, Natural Question test and Trivia QA test. The best performing models are marked **bold** and the results unavailable are left blank. The underlined coCondenser is reproduced by Zhang et al. (2022).

## 4.4 COLLAPSE ANALYSIS

In this part, we study the collapse problem that the multi-vector models may suffer from. We construct a new training set with an AVQ value of 1 by extracting the samples from NQ dataset. Then, we compare the retrieval performance between MVR and CAMVR using each view representation alone. As shown in Table 2, the performances of MVR model using each view representation and its overall accuracy are nearly identical. This indicates that all the view representations collapse into the same one. In contrast, the performances of each view representation in our CAMVR are distinguishable, and the overall performance is significantly better than that using each single view representation since the multiple view representations can complement to each other.

| NQ | AVQ=1, MVR | | | AVQ=1, CAMVR | | |
|---|---|---|---|---|---|---|
| | R@5 | R@20 | R@100 | R@5 | R@20 | R@100 |
| View1 | 55.62 | 73.19 | 84.35 | 32.55 | 49.70 | 66.87 |
| View2 | 55.51 | 73.21 | 84.38 | 59.28 | 75.37 | 84.82 |
| View3 | 55.40 | 73.32 | 84.43 | 58.92 | 73.96 | 84.24 |
| View4 | 55.40 | 73.16 | 84.35 | 59.17 | 73.66 | 83.74 |
| View5 | 55.60 | 73.16 | 84.32 | 58.09 | 73.24 | 83.63 |
| View6 | 55.54 | 73.21 | 84.32 | 57.73 | 72.33 | 81.44 |
| View7 | 55.37 | 73.21 | 84.32 | 55.40 | 71.69 | 83.02 |
| View8 | 55.60 | 73.19 | 84.32 | 56.90 | 72.58 | 83.35 |
| overall | 55.51 | 73.16 | 84.35 | 63.80 | 78.70 | 86.70 |

Table 2: Retrieval performance of each view on Nature Question test set for MVR and CAMVR under the case that AVQ value of training data is 1.

To learn multiple distinguishable view representations, attention probability vectors of views are expected to be different. This means different views should attend to different contextual information.

We further study the collapse problem by assessing the distance of each pair of attention vectors in the last layer of document encoder. The Kullack-Leibler divergence (KL divergence) is used as the distance metric. Let $\alpha_i = (\alpha_{i1}, \alpha_{i2}, \ldots, \alpha_{iL})$ be the attention probability vector of $[VIE_i]$. The KL divergence between $\alpha_i$ and $\alpha_j$ is defined as follows:

$$D_{\mathrm{KL}}(i,j) = \sum_{l=1}^{L} \alpha_{jl} \log \frac{\alpha_{jl}}{\alpha_{il}}, \tag{14}$$

which satisfies that $D_{\mathrm{KL}}(i,j) \geq 0$, and $D_{\mathrm{KL}}(i,j) = 0$ if and only if $\alpha_i = \alpha_j$. We randomly select 1000 documents from the retrieval corpus to calculate the average KL divergence between each pair of attention vectors. The results are shown in Table 3. All averaged KL divergence values between viewer tokens are 0 in MVR, which indicates that all viewer tokens attend to each token with nearly the same weights. In contrast, all the averaged KL divergence values of our CAMVR are greater than 0, which indicates that the viewer tokens are able to attend to different contextual information.

| View | MVR | | | | | | | | CAMVR | | | | | | | |
|---|---|---|---|---|---|---|---|---|---|---|---|---|---|---|---|---|
| | 1 | 2 | 3 | 4 | 5 | 6 | 7 | 8 | 1 | 2 | 3 | 4 | 5 | 6 | 7 | 8 |
| 1 | - | 0.0 | 0.0 | 0.0 | 0.0 | 0.0 | 0.0 | 0.0 | - | 0.50 | 0.63 | 0.54 | 0.42 | 0.37 | 0.36 | 0.33 |
| 2 | - | - | 0.0 | 0.0 | 0.0 | 0.0 | 0.0 | 0.0 | - | - | 0.21 | 0.51 | 0.62 | 0.52 | 0.40 | 0.36 |
| 3 | - | - | - | 0.0 | 0.0 | 0.0 | 0.0 | 0.0 | - | - | - | 0.26 | 0.65 | 0.81 | 0.77 | 0.69 |
| 4 | - | - | - | - | 0.0 | 0.0 | 0.0 | 0.0 | - | - | - | - | 0.25 | 0.59 | 0.72 | 0.64 |
| 5 | - | - | - | - | - | 0.0 | 0.0 | 0.0 | - | - | - | - | - | 0.20 | 0.41 | 0.42 |
| 6 | - | - | - | - | - | - | 0.0 | 0.0 | - | - | - | - | - | - | 0.12 | 0.20 |
| 7 | - | - | - | - | - | - | - | 0.0 | - | - | - | - | - | - | - | 0.07 |
| 8 | - | - | - | - | - | - | - | - | - | - | - | - | - | - | - | - |

Table 3: The KL divergences between attention vectors of viewer tokens for MVR and CAMVR.

### 4.5 VIEW INTERPRETABILITY ANALYSIS

In this section, we conduct the view interpretability analysis of our CAMVR. Unlike the MVR model, the views are not interpretable because each answer view is selected through the max-pooler operation in training shown as Figure 2(c). For CAMVR, we directly specify the answer view of positive document for each $(\boldsymbol{q}, \boldsymbol{d}^+)$ pair to supervise the learning process, which would force each view representation to capture more information of the corresponding snippet. To analyse the interpretability of each view, we compare the both overall and the retrieval accuracy exactly achieved by the answer view. For a given query $q_i$, we denote the corresponding top $K$ retrieved documents as $[d_1, \ldots, d_K]$, and their corresponding matched snippets are $\hat{S}_K = [ms_1, \ldots, ms_K]$, then the answer view accuracy is defined as:

$$\mathrm{ACC}_{ans}^{K} = \frac{1}{N_q} \sum_{i}^{N_q} I(\text{at least one snippet in } \hat{S}_K \text{ that contains the answer of } q_i), \tag{15}$$

where $N_q$ is the number of the test queries, $I(\cdot)$ is the indicator function. As shown in Table 4, the gaps between overall performance and answer view performance given the three datasets on top 5/20/100 are small, which indicates that the answer view representation successfully captures the information of the corresponding snippet, in which the scattered answers are located at.

| Method | SQuAD | | | Natural Question | | | Trivia QA | | |
|---|---|---|---|---|---|---|---|---|---|
| | R@5 | R@20 | R@100 | R@5 | R@20 | R@100 | R@5 | R@20 | R@100 |
| Overall | 69.0 | 79.2 | 86.0 | 71.1 | 82.5 | 89.1 | 77.3 | 84.1 | 87.9 |
| Answer View | 59.4 | 70.5 | 78.7 | 60.1 | 73.6 | 82.2 | 70.9 | 80.0 | 85.4 |
| Other Views | 9.6 | 8.7 | 7.3 | 10.0 | 8.9 | 6.9 | 6.4 | 4.1 | 2.5 |

Table 4: The comparison of overall performance and answer view performance for CAMVR on SQuAD, Natural Question and Trivia QA datasets.

To further illustrate the interetability of view representations, we take an example from NQ dataset for retrieval evaluation using MVR and CAMVR. As shown in Figure 4, answers of two queries are located at the 3rd and 5th snippets, respectively. For MVR model, the 3rd view is selected for the two queries in terms of time view and number view, which indicates MVR cannot distinguish the two view representations. In contrast, our CAMVR is able to answer the two queries, where the 3rd and 5th answer views are successfully selected.

**Title:** [VIE$_1$] Argentina at the FIFA World Cup
**Document:** [VIE$_2$] This is a record of Argentina's results at the FIFA World Cup. [VIE$_3$] Argentina is one of the most successful national football teams in the world, having won two World Cups in 1978 and 1986. [VIE$_4$] Argentina has been runners up three times in 1930, 1990 and 2014. [VIE$_5$] The team was present in all but four of the World Cups, being behind only Brazil, Italy and Germany in number of appearance. [VIE$_6$] Argentina has also won the Copa America 14 times, one less than Uruguay. [VIE$_7$] Moreover, Argentina has won the Confederations Cup and the gold medal. [VIE$_8$]

**Query1:** When was Argentina won the FIFA world cup
**Answer1:** 1978
**Query2:** Italy has won the world cup how many times
**Answer2:** Four

| | query | view1 | view2 | view3 | view4 | view5 | view6 | view7 | view8 |
|---|---|---|---|---|---|---|---|---|---|
| MVR | query1 | 56.76 | 56.73 | **56.77** | 56.76 | 56.75 | 56.74 | 56.73 | 56.72 |
| | query2 | 54.45 | 54.56 | **54.58** | 54.48 | 54.46 | 54.45 | 54.44 | 54.43 |
| CAMVR | query1 | 134.11 | 133.29 | **134.59** | 133.24 | 133.03 | 132.40 | 131.86 | 19.26 |
| | query2 | 132.12 | 132.44 | 132.38 | 132.24 | **132.96** | 131.61 | 131.17 | 21.59 |

Figure 4: Example of the document retrieved by MVR and CAMVR. Correct answers and selected views are written in **bold**.

## 4.6 VIEW NUMBER ANALYSIS

We conduct experiments to test the impact of view number on retrieval performance given the three datasets. The results with different view number varying from 1 to 12 are shown in Table 5. When the view number increases from 1 to 8, the performance is greatly improved. In particular, the improvements of top5 accuracy on the datasets of SQuAD, Natural Question and Trivia QA are up to 7.7, 2.7 and 1.8 points. The reason is that the average number of sentences per document is 6.5 in retrieval corpus, and 1 view or 4 views are insufficient to capture the fine-grained information of all snippets. The retrieval performance improved slightly when the view number is increased from 8 to 12 since the snippets information answering multi-view queries has almost been captured by the first 8 views.

| CAMVR | SQuAD | | | Natural Question | | | Trivia QA | | |
|---|---|---|---|---|---|---|---|---|---|
| | R@5 | R@20 | R@100 | R@5 | R@20 | R@100 | R@5 | R@20 | R@100 |
| n=1 | 61.3 | 74.5 | 83.4 | 68.4 | 80.8 | 87.8 | 75.5 | 82.9 | 86.0 |
| n=4 | 65.2 | 76.7 | 84.8 | 70.2 | 82.4 | 88.7 | 76.7 | 83.5 | 87.8 |
| n=8 | 69.0 | 79.2 | 86.0 | 71.1 | 82.5 | 89.1 | 77.3 | 84.1 | 87.9 |
| n=12 | 69.0 | 79.2 | 86.0 | 71.9 | 83.0 | 89.1 | 77.8 | 84.2 | 87.9 |

Table 5: Performance of different view number in CAMVR.

## 5 CONCLUSION

In this paper, we propose a context-adaptive multi-view representation learning framework for dense retrieval. Different from MVR model, where the representations would collapse into the same one with a low percentage of the documents answering multiple queries in training data, our proposed CAMVR learning framework is able to avoid the collapse problem by adaptively aligning each viewer token with each document snippet. In addition, we supervise the learning process by specifying the answer snippets for positive documents and allowing the representations to attend to local snippets, this additionally provides the interpretability for each view representation. The experimental results show that our proposed CAMVR is able to avoid the collapse problem and achieves the state-of-the-art retrieval performance.

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

## A  PERFORMANCE SIGNIFICANCE

To demonstrate the significant difference of the retrieval performance between CAMVR and MVR, we implement the two models based on pre-trained BERT and coCondenser given three datasets (SQuAD, Natural Question and Trivia QA) with 5 independent runs for each experimental setting. Due to the constraints of time and computational resources, all models are trained without utilizing mined hard negatives strategy. The experimental results are shown in Table 6. It is clear that the top5 accuracy of CAMVR is significantly better than MVR with a significance level of 0.05, and the top20 and top100 accuracy are also better than or comparable to MVR.

| Method | SQuAD | | | Natural Question | | | Trivia QA | | |
|---|---|---|---|---|---|---|---|---|---|
| | R@5 | R@20 | R@100 | R@5 | R@20 | R@100 | R@5 | R@20 | R@100 |
| MVR (coCondenser) | 59.1 | 73.0 | 82.8 | 70.0 | 82.5 | 89.3 | 76.4 | 83.5 | 87.7 |
| CAMVR (coCondenser) | 69.0* | 79.2* | 86.0* | 71.1* | 82.5 | 89.1 | 77.3* | 84.1* | 87.9 |
| MVR (BERT) | 61.4 | 73.2 | 82.5 | 66.2 | 79.0 | 86.6 | 72.8 | 81.6 | 86.5 |
| CAMVR (BERT) | 63.0* | 74.4* | 83.1* | 67.6* | 80.7* | 87.9* | 74.5* | 82.1* | 86.9* |

Table 6: The performance comparison of MVR and CAMVR based on pre-trained coCondenser and BERT. The asterisk indicates that the result is significant with a significance level of 0.05, according to a standard t-test.

## B  VIEW CAPACITY ANALYSIS

To evaluate the view capacity of capturing global information, we investigate the retrieval performance with each view as the document representation alone. The results shown in Table 7 demonstrate that each view of CAMVR can also capture the document-level information. Specifically, the top20 accuracy of view1, view2, view3 and view5 are comparable to the coCondenser (single-vector model), and surpass DPR by at least 9.8 points.

|         |         | Natural Question | | |
|---------|---------|------|-------|--------|
|         |         | R@5  | R@20  | R@100  |
| DPR     |         | -    | 74.4  | 85.3   |
| coCondenser | |     | 75.4 | 84.1  | 88.8   |
| CAMVR   | View1   | 75.1 | 84.3  | 89.5   |
|         | View2   | 74.3 | 84.2  | 89.6   |
|         | View3   | 75.1 | 84.2  | 89.4   |
|         | View4   | 75.2 | 84.0  | 89.5   |
|         | View5   | 74.7 | 84.2  | 89.1   |
|         | View6   | 73.4 | 83.5  | 88.9   |
|         | View7   | 71.9 | 82.7  | 88.8   |
|         | View8   | 68.0 | 79.8  | 87.6   |
|         | Overall | 76.6 | 85.1  | 89.9   |

Table 7: The view performance comparison between CAMVR and single-vector models.

## C  HARD NEGATIVES ANALYSIS

Since negative sampling is crucial to improve retrieval performance demonstrated by the existing literatures, we conduct the experiments to understand how the hard negatives improve the retrieval performance. To reduce the computational costs in negative sampling, we adopt the two-round training pipeline as coCondenser (Gao & Callan, 2021a). We firstly train a model by using the negative samples obtained via BM25, then the trained model is used to sample hard negatives for each query. Next, we retrain a model with the augmented training data containing the BM25 negatives and hard negatives. As shown in Table 8, all retrieval performances of the given three datasets are improved significantly with the hard negatives, especially for the top5 accuracy.

| Method | SQuAD | | | Natural Question | | | Trivia QA | | |
|--------|------|------|-------|------|------|-------|------|------|-------|
|        | R@5  | R@20 | R@100 | R@5  | R@20 | R@100 | R@5  | R@20 | R@100 |
| without HN | 69.0 | 79.2 | 86.0 | 71.1 | 82.5 | 89.1 | 77.3 | 84.1 | 87.9 |
| with HN | 77.1 | 84.5 | 88.6 | 76.6 | 85.1 | 89.9 | 80.1 | 85.5 | 88.8 |

Table 8: HN (hard negatives) ablation on CAMVR.

