# OpenReview forum: "CAMVR: Context-Adaptive Multi-View Representation Learning for Dense Retrieval"
_ICLR.cc/2023/Conference — Submitted to ICLR 2023_

### Official Review · Reviewer_tgtb · 2022-10-24

**Confidence:** 5
**Correctness:** 2
**Technical Novelty And Significance:** 2
**Empirical Novelty And Significance:** 2
**Recommendation:** 3

**Clarity, Quality, Novelty And Reproducibility:**

* For clarity, the paper is well-written and easy-to-follow.
* For novelty and technical quality, the proposed idea is not novel, and the experiments do not have enough technical quality to validate the proposed idea and have fair comparisons.
* As mentioned in weakness, the paper does not provide the information about reproducibility.


**Strength And Weaknesses:**

Strengths
* Important research topic.
* Publicly available datasets.
* Well-written and easy-to-follow.


Weaknesses
* The idea of aligning special tokens along with different parts in the document is not novel in the retrieval field. For example, [a] aligns special tokens along with different sentences. The authors should also consider comparing this line of existing studies.
* The semantic meaning of viewer tokens can become different after using the proposed arrangement. There may not be different viewer tokens for a specific span or at fair positions attending all documents.
* Following the previous point, many of the experiments and analysis are unfair to the original MVR because it has no such capability of doing that thing. The authors should compare valid methods like [a] for fair experiments.

[a] Jiang, J. Y., Xiong, C., Lee, C. J., & Wang, W. (2020, November). Long Document Ranking with Query-Directed Sparse Transformer. In Findings of the Association for Computational Linguistics: EMNLP 2020 (pp. 4594-4605).

**Summary Of The Paper:**

In this paper, the authors improve multi-view representation learning by splitting documents and aligning viewer tokens along with different parts. The experiments conducted on three datasets demonstrate that the proposed CAMVR performs better than MVR. The analysis also shows that CAMVR has better interpretability.

**Summary Of The Review:**

In sum, I would recommend “3: reject, not good enough” because the paper is not novel, and has some flaws in the experiments.

---

> ### Author Response · Authors · 2022-11-16
> **Response to Reviewer tgtb**
>
> **Q1: The idea of aligning special tokens along with different parts in the document is not novel in the retrieval field. For example, [a] aligns special tokens along with different sentences. The authors should also consider comparing this line of existing studies.**
>
>
> The main contributions are described in the general responses to the questions regarding the innovations of our paper.
>
> The idea of aligning special tokens along with different parts in the document is intuitive, our key contributions can be summarized as follows:
> 1) We essentially reveal and address the collapse problem of multi-view representations.
> 2) Our CAMVR introduces the interpretability for multi-view representations.
>
> Though the idea of aligning special tokens along with different parts in the document is used in [a] and our paper, the motivations are different as follows:
> 1) In [a], the idea is used to implement an efficient cross encoder by a sparse attention mechanism;
> 2) In our paper, we aim to produce multi-view embeddings (each view representation mainly captures the information of corresponding snippet) to represent documents and answer different queries.
>
> We have cited [a] in our paper in section 2.
>
> We don't compare our CAMVR with the QDS-transformer in [a] because of the following reasons:
> 1) QDS-transformer in [a] is a cross encoder model, where the query and document are concatenated into one text to accomplish a full interaction. Differently, our CAMVR is a dual encoder model, where the query and document are encoded separately. Cross encoder and dual encoder are two different lines of research in dense retrieval.
> 2) Rankers based on cross encoder architecture are computationally expensive for inference because of their cross-attention operations, so they are impractical for the first-stage retrieval given large-scale corpus. On the contrary, dual encoder is the widely adopted architecture because it can be easily and efficiently employed with the supports from approximate nearest neighbor (ANN).
>
>
> **Q2: The semantic meaning of viewer tokens can become different after using the proposed arrangement. There may not be different viewer tokens for a specific span or at fair positions attending all documents.**
>
> In our paper, the views of a document are to answer different queries. Each view can not only capture the information of the corresponding snippet (local information) but also the information of the entire document (global information).
>
> For the capacity of capturing the local information, as shown in Table 4, the gaps between overall performance and answer view performance given the three datasets on top5/20/100 are small, which indicates that the answer view representation successfully captures the information of the corresponding snippets.
>
> As supplementary, to evaluate the capacity of capturing global information, we investigate the retrieval performance with each view as the document representation alone. The results in the table below (Table 7) demonstrate that each view can also capture the document-level information. Specifically, the top20 retrieval accuracy of view1, view2, view3 and view5 are comparable to the coCondenser (single-vector model), and surpass DPR by at least 9.8 points.
>
> |Method|view|  | NQ|    |
> |------|----|---|----------------|---|
> |      |    |R@5|	R@20|	R@100|
> |DPR|CLS|-|	74.4|	85.3|
> |coCondenser|CLS|	75.4|	84.1|	88.8|
> |   |View1|	75.1|	84.3|	89.5|
> |	|View2|	74.3|	84.2|	89.6|
> |	|View3|	75.1|	84.2|	89.4|
> |	|View4|	75.2|	84.0|	89.5|
> |CAMVR|View5|	74.7	|84.2	|89.1|
> |	|View6|	73.4|	83.5|	88.9|
> |	|View7|	71.9|	82.7|	88.8|
> |	|View8|	68.0|	79.8|	87.6|
> |	|overall|	76.6|	85.1|	89.9|

---

> > ### Author Response · Authors · 2022-11-16
> > **Response to Reviewer tgtb**
> >
> >
> > **Q3: Following the previous point, many of the experiments and analysis are unfair to the original MVR because it has no such capability of doing that thing. The authors should compare valid methods like [a] for fair experiments.**
> > **[a] Jiang, J. Y., Xiong, C., Lee, C. J., & Wang, W. (2020, November). Long Document Ranking with Query-Directed Sparse Transformer. In Findings of the Association for Computational Linguistics: EMNLP 2020 (pp. 4594-4605).**
> >
> > We clarify all the experimental intentions and the choice of baseline models.
> >
> > 1) In Table 1, we mainly compare the retrieval performance of CAMVR with the existing dual encoder models. The existing baseline models are also not compared with cross encoder models because the cross encoders are computational intensive, such as QDS-transformer in [a].
> >
> > 2) In Table2 and Table3, we study the concerned collapse problem of multi-view representations by evaluating the retrieval performance of each view and the distance of each pair of view attention vectors. We have verified that the proposed context-adaptive mechanism is able to avoid the collapse problem.
> >
> > 3) In Table 4, we study the interpretability (not provided in the existing multi-view representation models) of each view representation introduced by our CAMVR model by comparing the overall performance and answer view performance. The results show the answer view representation is able to successfully capture the information of the corresponding snippet, which indicates the interpretability of all view representations.
> >
> > 4) In Table 5, we study the impact of the view number on the retrieval performance of our CAMVR model. The results show that the model performance improves greatly as the view number increases from 1 to 8. The reason is that the average number of sentences per document is 6.5 in retrieval corpus, and 1 view or 4 views are insufficient to capture the fine-grained information of all snippets.

---

### Official Review · Reviewer_b6Pg · 2022-10-25

**Confidence:** 4
**Correctness:** 3
**Technical Novelty And Significance:** 2
**Empirical Novelty And Significance:** 2
**Recommendation:** 5

**Clarity, Quality, Novelty And Reproducibility:**

**Clarity/Quality**
* Paper’s clarity and quality are in doubt. Paper has multiple typos and sections that require substantial polishing. For example, (13) does have a floating “i” variable and the connection to the per-view loss to the full loss function is not described. (7) has a form of the main loss function, but it is in the “Preliminary” section. Figure 4 has a typo. RAMVR->CAMVR. I encourage authors to improve the paper's clarity further.
* Even though snippet splitting can play a critical role in CAMVR, authors did not disclose information regarding the split. Authors just meant the split was by NLTK toolkit, but I am not sure how they exactly splitted. Please provide details.
* Important experiment details are missing. Authors should disclose models that they used in Section 4. (e.g. whether the model was pretrained, size, etc). I feel that the paper was rushed for the submission.

**Novelty**
CAMVR is a direct extension of MVR paper [1]. It essentially adds two minor contributions 1) inserting view tokens in between text segments unlike MVR, 2) adding the diversity loss function to prevent all multi-view representations to collapse. Both of the techniques do feel a minor addition to the original MVR paper.

[1] Zhang, Shunyu, et al. "Multi-View Document Representation Learning for Open-Domain Dense Retrieval." arXiv preprint arXiv:2203.08372 (2022).



**Strength And Weaknesses:**

**Strength**
* Simple and practical approach.
* Provides multiple analysis to examine the multi-views indeed are not collapsing.

**Weakness**
* Novelty could be questioned for some reviewers because it does feel it’s quite a simple extension of MVR paper.
* There is no principled way of splitting the text even though splitting can play a critical role in the proposal. Input text was arbitrarily split with a toolkit (NLTK) and it does not seem to be based on solid assumptions.
* Table 5 shows using 4 representations does not outperform other single-representation models (e.g. RocketQA/DPR-PAQ), which is a bit surprising. Only by using more than 8, it was able to outperform. Any discussions?


**Summary Of The Paper:**

This paper proposes to improve information retrieval models by introducing a new way of multi-representation learning, CAMVR. CAMVR extends MVR approach that added multi-view tokens at the beginning of the text inputs; instead, CAMVR adds the multi-view tokens in between the segments of text inputs, encouraging each multi-view representation to capture local information. Combined with the new loss function to prevent the collapsing multi-view representation problem, CAMVR were able to provide diverse document representations and demonstrated good performance in the common IR benchmark datasets.


**Summary Of The Review:**

As discussed above, the paper provides a simple practical approach in IR. However, the novelty is thin and paper does require further polishing.

---

> ### Author Response · Authors · 2022-11-16
> **Response to Reviewer b6Pg**
>
> **Q1: Novelty could be questioned for some reviewers because it does feel it’s quite a simple extension of MVR paper.**
>
> The main contributions are described in the general responses to the questions regarding the innovations of our paper.
>
>
> **Q2: There is no principled way of splitting the text even though splitting can play a critical role in the proposal. Input text was arbitrarily split with a toolkit (NLTK) and it does not seem to be based on solid assumptions.**
>
> In our paper, we split a document into multiple snippets according to the following principles:
> 1.	To guarantee the completeness of sentences, a sentence cannot be split into multiple snippets to avoid breaking the semantics. Thus, we use the sent_tokenize function in NLTK toolkit to split the document into sentences.
> 2.	To assign a relatively balanced length of a snippet to each viewer token so that the views are able to answer some queries, we merge two short adjacent sentences into one snippet.
>
> The details of document splitting are as follows:
>
> > **Algorithm 1** Document Splitting
> > 1. Input: A document $d$.
> > 2. Parameter: View number $n$ and empty sentence $b$.
> > 3. Output: A sequence $S$ containing $n$ snippets.
> >> 1. Split $d$ into $k$ sentences $S=[s_1,s_2,...,s_k]$ by the sent_tokenize function in NLTK tookit.
> >> 2. if $k\le n$ then
> >> +  Add $n-k$ empty sentences $b$ to $S$ and obtain the final snippets $S=[s_1,s_2,...,s_k]$.
> >> +  else
> >>>  while $len(S)>n$ do
> >>> + Identify the shortest sentence $s_i$ in $S$.
> >>> + Merge $s_i$ with its shorter adjacent sentence $s_j$ into a new snippet $s'_i$.
> >>> + $s_i \gets s'_i$
> >>> + Remove $s_j$ from $S$.
> >>> + end while
> >> + end if
>
>
> **Q3: Table 5 shows using 4 representations does not outperform other single-representation models (e.g. RocketQA/DPR-PAQ), which is a bit surprising. Only by using more than 8, it was able to outperform. Any discussions?**
>
> We thank the reviewer for the insightful suggestion. We found that the experimental results with a view number of 4 were indeed questionable. Therefore, we double-check the parameter settings and re-run the experiments. As complementary, we also report the retrieval performance when the view number is 1 in Table 5. As shown in the table below, the overall performance improves with the increase of view number.
>
> |CAMVR|    |SQuAD|	 |   |NQ|	  |  |TQA|  |
> |-----|----|-----|---|---|--------------|----|---|------|---|
> |	|R@5|	R@20|	R@100|	R@5|	R@20|	R@100|	R@5|	R@20|	R@100|
> |n=1|	61.3|	74.5|	83.4|	68.4|	80.8|	87.8|	75.5|	82.9|	86|
> |n=4|	65.2|	76.7|	84.8|	70.2|	82.4|	88.7|	76.7|	83.5|	87.8|
> |n=8|	69.0|	79.2|	86.0|	71.1|	82.5|	89.1|	77.3|	84.1|	87.9|
> |n=12|	69.0|	79.2|	86.0|	71.9|	83.0|	89.1|	77.8|	84.2|	87.9|
>
> **Q4: Paper’s clarity and quality are in doubt. Paper has multiple typos and sections that require substantial polishing. For example, (13) does have a floating “i” variable and the connection to the per-view loss to the full loss function is not described. (7) has a form of the main loss function, but it is in the “Preliminary” section. Figure 4 has a typo. RAMVR->CAMVR. I encourage authors to improve the paper's clarity further.**
>
> We thank the reviewer for the question. All the typos have been corrected in the new submitted paper.
>
> We are sorry for the confusing description of equation (13)
> $$
> \begin{align}
>     \mathcal{L}=-\log\frac {\exp(f_i(q,d^+)/\tau)}{\exp(f_i(q, d^+)/\tau) + \sum_l \exp(f(q, d_l^-)/\tau)},
> \end{align}
> $$
> It is the loss function of CAMVR. Since we specify the answer view in our supervised learning process and set the i-th view as the answer view, $f_i(q, d^+)$ is used as the similarity score between a query $q$ and positive document $d^+$ given the answer view.

---

> > ### Author Response · Authors · 2022-11-16
> > **Response to Reviewer b6Pg**
> >
> > **Q5: Even though snippet splitting can play a critical role in CAMVR, authors did not disclose information regarding the split. Authors just meant the split was by NLTK toolkit, but I am not sure how they exactly splitted. Please provide details.**
> >
> > In our paper, we split document into multiple snippets according to the following principles:
> > 1.	To guarantee the completeness of sentences, a sentence cannot be split into multiple snippets to avoid breaking the semantics. Thus, we use the sent_tokenize function in NLTK to split the document into sentences.
> > 2.	To assign a relatively balanced length of a snippet to each viewer token so that the views are able to answer some queries, we merge two short adjacent sentences into one snippet.
> >
> > The details of document splitting are as follows:
> >
> > > **Algorithm 1** Document Splitting
> > > 1. Input: A document $d$.
> > > 2. Parameter: View number $n$ and empty sentence $b$.
> > > 3. Output: A sequence $S$ containing $n$ snippets.
> > >> 1. Split $d$ into $k$ sentences $S=[s_1,s_2,...,s_k]$ by the sent_tokenize function in NLTK tookit.
> > >> 2. if $k\le n$ then
> > >> +  Add $n-k$ empty sentences $b$ to $S$ and obtain the final snippets $S=[s_1,s_2,...,s_k]$.
> > >> +  else
> > >>>  while $len(S)>n$ do
> > >>> + Identify the shortest sentence $s_i$ in $S$.
> > >>> + Merge $s_i$ with its shorter adjacent sentence $s_j$ into a new snippet $s'_i$.
> > >>> + $s_i \gets s'_i$
> > >>> + Remove $s_j$ from $S$.
> > >>> + end while
> > >> + end if
> >
> >
> > **Q6: Important experiment details are missing. Authors should disclose models that they used in Section 4. (e.g. whether the model was pre-trained, size, etc). I feel that the paper was rushed for the submission.**
> >
> > We thank the reviewer for the question.	In this paper, we use the coCondenser pre-trained model as the base model without any further pretraining. The training details are presented in Section 4.2.
> >
> >
> > **Q7: CAMVR is a direct extension of MVR paper [1]. It essentially adds two minor contributions 1) inserting view tokens in between text segments unlike MVR, 2) adding the diversity loss function to prevent all multi-view representations to collapse. Both of the techniques do feel a minor addition to the original MVR paper.**
> >
> > The main contributions are described in the general responses to the questions regarding the innovations of our paper.
> >
> > To clarify the proposed “diversity loss function”, we specify the answer view to supervise the representation learning process, which makes each view attends more to its corresponding snippet, thus the interpretability of view representations can be given.

---

### Official Review · Reviewer_dw1C · 2022-10-25

**Confidence:** 4
**Correctness:** 3
**Technical Novelty And Significance:** 3
**Empirical Novelty And Significance:** 2
**Recommendation:** 5

**Clarity, Quality, Novelty And Reproducibility:**

This paper is well motivated to learn distinguished representations for multi-views. The diagrams explicitly illustrate the difference between the proposed method and prior methods in terms of the model architecture.

The idea of aligning each viewer token with different document snippets is intuitive. However, given the prior work MVR (Zhang et al., 2022), the novelty of this paper is thin. The method proposed in this paper seems incremental to existing work.

The paper discusses how the parameters and hyper-parameters are set, hence the results should be reproducible.

**Strength And Weaknesses:**

Strength

1. This paper focuses on an interesting and impactful problem, that is the multi-view representations tend to collapse into the same one when the percentage of documents answering multiple queries in training data is low. Resolving this problem is critical to reading comprehension and question answering tasks.

2. The proposed context-adaptive multi-view representation model is reasonable and empirically effective.

Weakness:

1. Missing technical details:
(1) Are the results, for example, in Table 1 statistically significant? It'd be great t-test results could be presented, in order to make the conclusion of this paper more convincing.
(2) It seems the proposed method introduces a hyper-parameter, that is the number of snippets to divide a document. More discussions would be interesting.


**Summary Of The Paper:**

This paper aim at overcoming the drawback of multi-view representation models for dense-vector based open domain retrieval or question answering. A context-adaptive multi-view representation learning framework is proposed to avoid the collapse of representations in different views by adaptively aligning each viewer token with each document snippet. The answer snippets for positive documents are specified in a supervised-learning setting and the representations are able to attend to local snippets, this additionally enhances the interpretability for each view representation. The experimental results on multiple reading comprehension and question answering datasets show that the proposed approach outperforms state-of-the-art methods.





**Summary Of The Review:**

This paper studies the collapse problem that multi-vector models have in learning identical representations in different views. A simple and effective method is proposed to learn multi-view representations that attend to the local snippets of a document.

In general, the proposed method is well-motivated and reasonable. However, the paper has not presented critical technical details, including how significant are the results statistically and how exactly a document is divided into multiple short snippets by NLTK toolkit.

The contribution of this paper is incremental given prior works have presented many insights such as how to apply hard negatives.

---

> ### Author Response · Authors · 2022-11-16
> **Response to Reviewer dw1c**
>
> **Q1: Are the results, for example, in Table 1 statistically significant? It'd be great t-test results could be presented, in order to make the conclusion of this paper more convincing.**
>
> To demonstrate the significant difference of the retrieval performance between CAMVR and MVR, we implemented the two models based on BERT and coCondenser pre-trained models given three datasets (SQuAD, NQ and TriviaQA) with 5 independent runs for each experimental setting.
>
> Due to the constraints of time and computational resources, all models are trained without utilizing the hard negatives strategy. The experimental results are shown in the following table (Table 6 in Appendix A). It is clear that the top5 accuracy of CAMVR is significantly better than that of MVR with a significance level of 0.05, and the top20 and top100 accuracy are also better than or comparable to MVR.
>
> |   | 	|SQuAD|   |	    |NQ|	|   |TQA|    |
> |---|---|-----|------|-----|--------|--------|-----|--------|--------|
> |	|R@5|R@20 |	R@100|	R@5|	R@20|	R@100|	R@5|	R@20|	R@100|
> |MVR (coCondenser)|	59.1|	73.0|	82.8|	70|	82.5|	89.3|	76.4|	83.5|	87.7|
> |CAMVR (coCondenser)|	69*|	79.2*|	86*|	71.1*|	82.5|	89.1|	77.3*|	84.1*|	87.9|
> |MVR (BERT)|	61.4|	73.2|	82.5|	66.2|	79|	86.6|	72.8|	81.6|	86.5|
> |CAMVR (BERT)|	63*|	74.4*|	83.1*|	67.6*|	80.7*|	87.9*|	74.5*|	82.1*|	86.9*|
>
> **Q2: It seems the proposed method introduces a hyper-parameter, that is the number of snippets to divide a document. More discussions would be interesting.**
>
> In our paper, the number of snippets is determined by the predefined view number and the discussion of view number has been presented in Table 5. Specifically, the snippets of a document are obtained by the following algorithm (Algorithm 1) :
>
> > **Algorithm 1** Document Splitting
> > 1. Input: A document $d$.
> > 2. Parameter: View number $n$ and empty sentence $b$.
> > 3. Output: A sequence $S$ containing $n$ snippets.
> >> 1. Split $d$ into $k$ sentences $S=[s_1,s_2,...,s_k]$ by the sent_tokenize function in NLTK toolkit tookit.
> >> 2. if $k\le n$ then
> >> +  Add $n-k$ empty sentences $b$ to $S$ and obtain the final snippets $S=[s_1,s_2,...,s_k]$.
> >> +  else
> >>>  while $len(S)>n$ do
> >>> + Identify the shortest sentence $s_i$ in $S$.
> >>> + Merge $s_i$ with its shorter adjacent sentence $s_j$ into a new snippet $s'_i$.
> >>> + $s_i \gets s'_i$
> >>> + Remove $s_j$ from $S$.
> >>> + end while
> >> + end if
>
>
> **Q3: The idea of aligning each viewer token with different document snippets is intuitive. However, given the prior work MVR (Zhang et al., 2022), the novelty of this paper is thin. The method proposed in this paper seems incremental to existing work.**
>
> The main contributions are described in the general responses to the questions regarding the innovations of our paper.
>
> **Q4: The contribution of this paper is incremental given prior works have presented many insights such as how to apply hard negatives.**
>
> We use the hard negatives just for a fair comparison with the existing methods. Considering that the answers of different queries are located in the different snippets of a document, the view to answer a query should capture the information of the snippet containing the answer and perceive the document-level information. Thus, we propose our CAMVR method and the answer view supervision mechanism. In addition, the theoretical analysis and extensive experimental results confirm the superiority of our method.

---

### Author Response · Authors · 2022-11-16
**Response to all reviewers and AC**

Dear Reviewers and AC:

We sincerely appreciate your valuable time and constructive comments.

Regarding the questions about the novelty from three reviewers, we emphasize that there are two key contributions compared with the existing multi-vector models in dense retrieval:

(1)	Since a long document can usually answer multiple potential queries from different views, it is intuitive and reasonable to encode a document into multiple representations aligning with different queries. However, the existing methods, such as MVR (Zhang et al. 2022) and ME-BERT (Luan et al., 2020), suffer from a serious collapse problem, where the view representations of a document may collapse into the same one. Therefore, the queries from different views tend to be answered by the same view representation of a document.

In our paper, we essentially analyze the collapse problem of MVR and theoretically reveal that the collapse problem is mainly caused by the fixed positions of views. Then we propose the CAMVR learning framework to effectively avoid the collapse problem by adaptively aligning each viewer token with a specific document snippet. To the best of our knowledge, our paper is the first work that essentially reveals and addresses the collapse problem of multi-view representations in dense retrieval.

(2)	Our paper is also the first work to provide the interpretability of the multi-view representations, which is crucial to explicitly analyze the retrieval process for a long document. In the existing multi-vector models, a max-pooler is usually adopted to select the answer views, this introduces indistinguishability and low interpretability of the multi-view representations of a document. In our CAMVR, we specify the answer view to supervise the representation learning process, which makes each view attend more to its corresponding snippet, thus the interpretability of view representations can be given. More detailed results are shown in Table 4.


Regarding the feedbacks from reviewers, we have made the following changes. All essential changes in the new version of paper are highlighted in blue color.
1. We added the performance significance experiments in Appendix A and verified that our CAMVR significantly outperforms MVR.
2. We added the document splitting algorithm in section 3.2 to describe the details of splitting a document into snippets.
3. We modified the experimental results in Table 5 by reconducting the experiments. Specifically, the retrieval performance when $n = 4$ is corrected and the retrieval performance when $n = 1$ is added.
4. We corrected the typo in Figure 4 (RAMVR to CAMVR).
5. We corrected the Equation (13) and explained why we use $f_i(q,d^+)$ as the similarity between the given query $q$ and positive document $d^+$.
6. We added the view capacity analysis in Appendix B. To assess the capacity of capturing global information of the views in CAMVR, we studied the retrieval performance by using each view as the document representation alone and demonstrated that each view in CAMVR is able to capture the document-level information.
7. We put the section 4.7 (hard negative analysis) to Appendix C.
8. According to the comments from Reviewer tgtb, we cited the paper [a] in section 2.
9. We modified section 4.2 to explicitly clarify that we use the pre-trained coCondenser as the base model without any further pre-training.

---

### Decision · Program_Chairs · 2023-01-20

**Decision:**

Reject

**Justification For Why Not Higher Score:**

While the problem is an important problem in the community, the contributions of the paper was not very new given what the previous work has presented with regard to applying hard negatives.

**Justification For Why Not Lower Score:**

N/A

**Metareview: Summary, Strengths And Weaknesses:**

Summary: The paper has a neat contribution that improves MVR but the reviewers were not excited about the paper, mostly due to weaknesses as mentioned below. While the authors were able to improve the paper further during the rebuttal, it is not clear if the paper will have enough interested audience in the conference.

Strengths:
- Most reviewers agree that the problem is interesting and important.
- Most reviewers agree that the proposed method is reasonable and effective in the given datasets.

Weaknesses:
- Most reviewers think that the proposed method is a relatively simple extension of previous work (e.g. MVR).
- Some reviewers think that comparing with MVR is not fair and there should be more appropriate baselines.
- Many reviewers are concerned that the paper is not clear enough.